# Fractal Patterns May Unravel the Intelligence in Next-Token Prediction

## Abstract

We study the fractal structure of language, aiming to provide a precise formalism for quantifying several properties that may have been previously suspected but not formally shown. We establish that language is: (1) *self-similar*, exhibiting complexities at all levels of granularity, with no particular characteristic granularity level or context length, and (2) *long-range dependent* (LRD), with tokens at any instant typically correlated with all subsequent tokens. Based on these findings, we argue that short-term patterns in language, such as in paragraphs, mirror the patterns seen in larger scopes, like entire documents. This may shed some light on how next-token prediction can lead to a comprehension of the structure of text at multiple levels of granularity, from words and clauses to broader contexts and intents. In addition, we demonstrate a connection between fractal parameters, such as the Hurst exponent, and scaling laws when varying the context length at inference time. We hope that these findings offer a fresh perspective on the nature of language and the mechanisms underlying the success of LLMs.

## 1 Introduction

How does next-token prediction in large language models (LLMs) yield remarkably intelligent behavior? Consider, for instance, the two models: GPT4 (OpenAI, 2023) and PaLM2 (Anil et al., 2023); these models have demonstrated extraordinary capabilities beyond just mastering language. Their skills extend to quantitative reasoning, creative content creation, document summarization, and even coding, which has prompted some researchers to ponder if there was more to intelligence than "on-the-fly improvisation" (Bubeck et al., 2023). While understanding the exceptional capabilities of LLMs is complex, particularly given the fuzzy meaning of "intelligent" behavior, a possible insight can be drawn from the study of fractals and self-similarity. We elucidate this connection in this work.

**Self-Similarity.** Self-similar processes were introduced by Kolmogorov in 1940 (Kolmogorov, 1940). The notion garnered considerable attention during the late 1960s, thanks to the extensive works of Mandelbrot and his peers (Embrechts & Maejima, 2000).

Broadly speaking, an object is called "self-similar" if it is invariant across scales, meaning its statistical or geometric properties stay consistent irrespective of the magnification applied to it (see Figure 1). Nature furnishes us with many such patterns, such as coastlines and snowflakes. In geometry, the Cantor set and the Kuch curve stand out as beautiful illustrations of this phenomenon. Despite the distinction, self-similarity is often discussed in the context of "fractals," another term popularized by Mandelbrot in his seminal book *The Fractal Geometry of Nature* (Mandelbrot, 1982). However, the two concepts are different (Gneiting & Schlather, 2004). We define each in Section 2.

In language, in particular, there have been studies arguing for the presence of a self-similar structure. Nevertheless, due to the computational constraints of the past, it was not feasible to holistically model the joint probability distribution of language. As such, linguists often resorted to rudimentary approximations in their arguments, such as by substituting a word with its frequency or length (Ausloos, 2012), or by focusing on the recurrence of a specific, predetermined word (Najafi & Darooneh, 2015; Altmann et al., 2012). These studies fall short of fully capturing the underlying structure of language due to the simplifying assumptions they make. For example, they remain invariant to the semantic ordering of texts and do not model second-order statistics, such as long-range dependence (LRD) (Najafi & Darooneh, 2015).

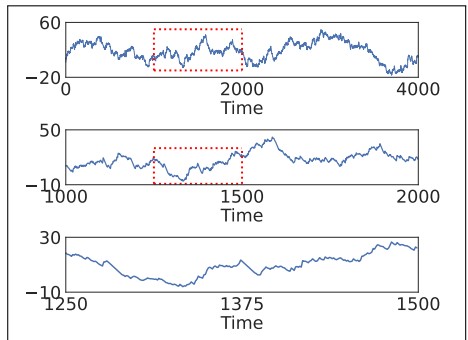 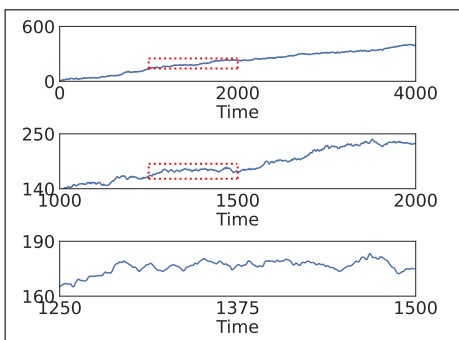

Figure 1: Manifestations of processes across different time scales. A region marked in red corresponds to the magnified plot shown below it. LEFT: The process exhibits self-similarity with rich details at all levels of granularity. It is an integral process $(X_t)_{t \in \mathbb{N}}$ calculated from Wikipedia (see Section 2). RIGHT: Example of a process that is not self-similar, looking smoother at larger time scales.

Highlighting the self-similar nature of a process can have profound implications. For instance, conventional Poisson models for Ethernet traffic were shown to fail because traffic was self-similar (Crovella & Bestavros, 1995; Leland et al., 1994; Paxson & Floyd, 1995; Willinger et al., 1997). In such cases, recognizing and quantifying this self-similarity had practical applications, such as in the design of buffers in network devices (Wilson, 2004). Similarly in language, we argue that self-similarity may offer a fresh perspective on the mechanisms underlying the success of LLMs. Consider the illustrative example shown in Figure 1 (left), where the task is to predict the subsequent observation in a time series, specifically predicting next tokens in a Wikipedia article (see Section 2 for details). The three plots in Figure 1 (left) represent different manifestations of the same process observed across three distinct time scales. Notably, we can observe rich details, e.g. burstiness, in *all* of them. Hence, for the model to successfully predict the next observation, it must capture the behavior of the process at various levels of granularity. The common approach for quantifying self-similarity is using the Hölder exponent (Watkins, 2019), which we denote by S. In language, we estimate it to be $S = 0.638 \pm .006$ (see Section 2 for details), confirming that the process exhibits statistical self-similarity.

Why is this significant? We hypothesize that since LLMs are trained to predict the future of a self-similar process, i.e., language, they develop proficiency in capturing behavior across multiple levels of granularity for two interconnected reasons: First, self-similarity implies that the patterns in language at the level of a paragraph are reflective of the patterns seen at the level of a whole text. Hence, recognizing short-term patterns can also aide in learning broader contexts. Second, because language displays detailed, intricate patterns at every level of granularity, it would not be enough to rely only on the immediate context of a sentence to predict the next token. Instead, the model would need to identify and predict patterns at higher levels of granularity; i.e. understand the overarching topic, direction of the argument, and even the broader context and intent. It must balance between immediate and long-term contexts. Willinger et al. (1995) and Altmann et al. (2012) argue that self-similarity might arise in language precisely because of this hierarchical nature.

**Long-range dependence.** However, self-similarity by itself is not sufficient for a predictive model to exhibit anything resembling "intelligent" behavior. In fact, some self-similar processes, despite their intricacy across all levels of granularity, remain entirely unpredictable. A quintessential example is the simple Brownian motion, which is a Wiener process with independent increments. Its discrete analog $B_n$ is defined by $B_n = \sum_{i=1}^{n} \varepsilon_i$, where $\varepsilon_i \sim \mathcal{N}(0, \sigma^2)$. Despite possessing rich details at all granularities, a model trained to predict the future of a simple Brownian motion cannot obviously acquire any intelligence since the process itself is completely unpredictable.

Thus, for intelligent behavior to manifest, the process must have some degree of predictability or *dependence* as well. One classical metric for quantifying predictability in a stochastic process is the Hurst parameter (Hurst, 1951), developed by the hydrologist H. E. Hurst in 1951 while studying the Nile river flooding. It is generally considered to be a robust metric (Willinger et al., 1995), unlike for instance the wavelet estimator (Abry et al., 1995) and the periodogram method (Geweke & Porter-Hudak, 1983) that can be sensitive to measurement errors (Pilgrim & Taylor, 2018). As we

discuss in Section 2, we estimate the Hurst parameter in language to be $H = 0.74 \pm 0.02$. For context, the Hurst parameter can only take values in $[0, 1]$. A higher value suggests more predictability or persistence in the data ($H = 1$ for completely deterministic system), while a lower Hurst parameter indicates more randomness ($H = 0.5$ for completely random system). See Section 2.3.

While it is compelling that $H \approx 0.75$ in language lies *midway* between determinism and noise, it is perhaps more surprising how similar that value is to what Hurst himself calculated for the Nile river, which is $H \approx 0.77$ (Hurst, 1951), and to what has been estimated for Ethernet traffic, which is $H \approx 0.75$ (Crovella & Bestavros, 1995). In fact, it turned out that a Hurst parameter of about $0.75$ occurs commonly in nature (Feller, 1951; Aref, 1998). Many processes, such as those relating to river discharges, temperatures, precipitation, and tree rings consistently exhibit similar values.

Importantly, predictability and self-similarity *together* imply long-range dependence (LRD). This follows from the definition of self-similarity, where the patterns at small scales mirror those at larger scales so, for example, the correlations established at micro levels are also pertinent at macro levels. LRD is arguably necessary for intelligence to emerge in a predictive model because processes that only exhibit short-range dependence could be forecasted (somewhat trivially) using lookup tables that provide the likelihood of transitions over brief sequences. By contrast, this is not possible in LRD processes due to the long contexts, which extend indefinitely into the past.

**Statement of Contribution.**    In summary, our contribution is to:

1. highlight how the fractal structure of language can offer a unique perspective on the intelligent behavior exhibited by large language models (LLMs), and provide a precise formalism to quantify properties of language, such as long-range dependence (LRD).

2. establish that language is self-similar and long-range dependent. We provide concrete estimates in language of the three parameters: the self-similarity (Hölder) exponent, the Hurst parameter, and the fractal dimension. We also estimate the related Joseph exponent.

3. demonstrate a connection between fractal patterns, such as the Hurst exponent, and scaling law parameters when varying the context length at inference time.

## 2    Fractal Parameters of Language

### 2.1    Preliminaries

Suppose we have a discrete-time, stationary stochastic process $(x_t)_{t \in \mathbb{N}}$. We assume that $\mathbb{E}[x_t] = 0$ and $\mathbb{E}[x_t^2] = 1$. We will refer to $(x_t)_{t \in \mathbb{N}}$ as the *increment process* to distinguish it from the *integral process* $(X_t)_{t \in \mathbb{N}}$ defined by $X_t = \sum_{k=0}^{t} x_k$. While $(x_t)_{t \in \mathbb{N}}$ and $(X_t)_{t \in \mathbb{N}}$ are merely different representations of the same data, it is useful to keep both representations in mind. For example, self-similarity is typically studied in the context of integral processes whereas long-range dependence (LRD) is defined on increment processes.

In the literature, it is not uncommon to mistakenly equate parameters that are generally different. For example, the Hurst parameter has had many different definitions in the past that were not equivalent, and Mandelbrot himself had cautioned against this (Mandelbrot, 2002). The reason behind this is because different parameters can agree in the idealized fractional Brownian motion setting, leading some researchers to equate them in general (Watkins, 2019). We will keep the self-similarity exponent S and the Hurst parameter H separate in our discussion.

**Experimental Setup.**    In order to establish self-similarity and LRD in language, we convert texts into sequences of bits using a language model (LM). Specifically, we use PaLM-8B (Chowdhery et al., 2022) to calculate the probability of the next word $w_t$ conditioned on its entire prefix $w_{[t-1]} = (w_0, w_1, \ldots, w_{t-1})$. By the chain rule (Cover, 1999), the corresponding number of bits assigned to $w_t$ is $z_t = -\log p(w_t | w_{[t-1]})$. Unlike in prior works, which rely on simplifications such as by substituting a word with its length (Ausloos, 2012) or by focusing on the recurrence of a single, predetermined word (Najafi & Darooneh, 2015; Altmann et al., 2012), we use the language model to approximate the full joint distribution of language. We carry out these calculations for prefixes of up to 4096 words (i.e. approximately 15 pages of text). The size of the PaLM model we use is 8 billion parameters, trained on approximately 780B tokens. See Chowdhery et al. (2022) for further details.

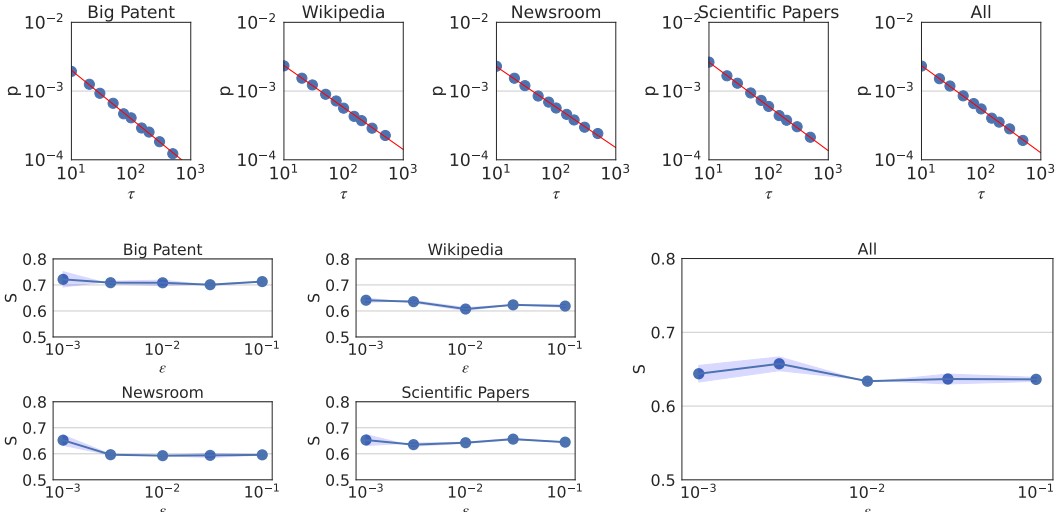

Figure 3: TOP: The peak probability $p_\epsilon(\tau)$ with $\epsilon = 10^{-2}$ is plotted against the granularity level $\tau$. See Section 2.2 for details. We observe a power law relation $p_\epsilon(\tau) \sim \tau^{-S}$ in all datasets, with exponents ranging from $S = 0.59 \pm 0.01$ in Newsroom to $S = 0.70 \pm 0.02$ in the Big Patent. When all datasets are aggregated (rightmost plot), we have $S = 0.63 \pm 0.01$. The existence of a power law relation indicates a self-similar structure, in agreement with Figure 1. BOTTOM: Sensitivity analysis of the self-similarity exponent to $\epsilon$. Generally, S is insensitive to the choice of $\epsilon$.

To rely on a language model for such analysis, it must provide probability scores that are reasonably well-calibrated. Generally, LLMs are known to produce calibrated probability scores at the token level (Kadavath et al., 2022). As a sanity check, we compare the logits $-\log p(\text{word})$ predicted by PaLM-8B with the actual log probabilities derived from the Google Web Trillion Word Corpus (Brants & Web, 2006) based on word frequencies. We use histogram binning (by grouping similar logits together) and plot their averaged actual log probabilities, similar to how the expected calibration error (ECE) is calculated (Guo et al., 2017). The results are presented in Figure 2. Notably, we find a strong agreement for the most frequently occurring words, i.e., when the word probability exceeds $p \gg 10^{-9}$.

Once $z_t$ is computed for a document, we construct the increment process $(x_t)_{t \in \mathbb{N}}$ by normalizing $z_t$ to have a zero-mean and unit variance. The integral process $(X_t)_{t \in \mathbb{N}}$ is calculated based on $(x_t)_{t \in \mathbb{N}}$, as described earlier and depicted in Figure 1 (left). Normalizing bits (to have zero mean and unit variance) models language as a random walk. It is a standard approach used extensively in the literature in various contexts, such as in DNA sequences (Peng et al., 1992; Roche et al., 2003; Montemurro & Pury, 2002; Kokol & Podgorelec, 2000; Schenkel et al., 1993).

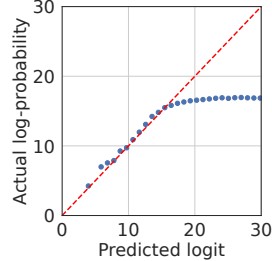

Figure 2: A comparison of PaLM-8B's logits with actual log-probabilities. We observe a substantial agreement except for exceedingly uncommon words with a probability less than $10^{-9}$.

We use four datasets all containing a minimum of 1,000 documents of length $> 4K$ words. They are: (1) Big Patent, which contains records of U.S. patent documents (Sharma et al., 2019), (2) Wikipedia, containing English articles only (Wikimedia, 2023), (3) Newsroom, which is also a collection of articles (Grusky et al., 2018), and (4) Scientific Papers, derived from ArXiv and PubMed OpenAccess repositories (Cohan et al., 2018). We restrict analysis to sufficiently-long documents of length $> 4K$ words. We use the bootstrap method (Efron & Tibshirani, 1994) to estimate the margin of error.

**Notation.** We write $f(x) \sim x^c$ if $f(x) = x^c L(x)$ for some slowly-varying function $L$. More precisely, $L(tx)/L(x) \to 1$ as $x \to \infty$ for all $t > 0$. Examples of slowly varying functions are constants $L(x) = c$ and logarithmic functions $L(x) = \log x$. When $f(x) \sim x^c$, we will abuse terminology slightly by referring to $f(x)$ as a power law function.

## 2.2 SELF-SIMILARITY EXPONENT

An integral process is said to be self-similar if it exhibits *statistical* self-similarity. More precisely, $(X_t)_{t \in \mathbb{N}}$ is self-similar if $(X_{\tau t})_{t \in \mathbb{N}}$ is distributionally equivalent to $(\tau^S X_t)_{t \in \mathbb{N}}$ for some exponent S. Thus, scaling of time is equivalent to an appropriate scaling of space. We will refer to $\tau$ as the *granularity level* and to the exponent S as the self-similarity exponent. It is worth noting that S is also often called the Hölder exponent (Watkins, 2019). Many time series in nature exhibit self-similar structures, such as human blood pressure and heart rate (Goldberger et al., 2002).

One convenient approach for calculating the self-similarity exponent S is as follows. First, fix $\epsilon \ll 1$ and denote the $\tau$-increments by $(X_{t+\tau} - X_t)_{t \in \mathbb{N}}$. These would correspond, for instance, to the number of bits used for clauses, sentences, paragraphs and longer texts as $\tau$ increases. In terms of the increment process $(x_t)_{t \in \mathbb{N}}$, this corresponds to aggregating increments into "bursts". Let $p_\epsilon(\tau)$ be the probability mass of the event $\{|X_{t+\tau} - X_t| \leq \epsilon\}_{t \in \mathbb{N}}$. Then, S can be estimated by fitting a power law relation $p_\epsilon(\tau) \sim \tau^{-S}$ (Watkins, 2019). We adopt this approach in our experiments.

Figure 3 (top) plots the probability $p_\epsilon(\tau)$ against $\tau$ when $\epsilon = 10^{-2}$. We indeed observe a power law relation; i.e. linear in a log-log scale. When all the datasets are aggregated, the self-similarity exponent is $S = 0.63 \pm 0.01$. Figure 3 (bottom) shows that the S is robust to the choice of $\epsilon$.

## 2.3 HURST PARAMETER

The Hurst parameter $H \in [0, 1]$ quantifies the degree of predictability or dependence over time (Hurst, 1951). It is calculated using the so-called rescaled-range (R/S) analysis. Let $(x_t)_{t \in \mathbb{N}}$ be an increment process. For each $n \in \mathbb{N}$, write $y_t = x_t - \frac{1}{t}\sum_{k=0}^{t} x_k$ and $Y_t = \sum_{k=0}^{t} y_t$. The range and scale are defined, respectively, as $R(n) = \max_{t \leq n} Y_t - \min_{t \leq n} Y_t$ and $S(n) = \sigma(\{x_k\}_{k \leq n})$, where $\sigma$ is the standard deviation.

Then, the Hurst parameter H is estimated by fitting a power law relation $R(n)/S(n) \sim n^H$. As stated earlier, for completely random processes, such as a simple Brownian motion, it can be shown that $H = 1/2$. On the other hand, $H = 1$ is a deterministic system. Hence, $H > 1/2$ implies dependence over time (Crovella & Bestavros, 1995; Willinger et al., 1995; Aref, 1998).

Writing $\rho_n = \mathbb{E}[(x_{t+n} x_t]$ for the autocovariance function of the increment process $(x_t)_{t \in \mathbb{N}}$, the Hurst parameter satisfies $H = 1 - \beta/2$ when $\rho_n \sim n^{-\beta}$ as $n \to \infty$ (Gneiting & Schlather, 2004; Crovella & Bestavros, 1995). Since in self-similar processes, $H > 1/2$ implies long-range dependence (LRD), LRD is equivalent to the condition that the autocovariances are not summable.

In terms of the integral process, it can be shown that (Samorodnitsky, 2006; Altmann et al., 2012):

$$\lim_{n \to \infty} \frac{\mathrm{Var}(X_n)}{n} = 1 + 2\sum_{i=1}^{\infty} \rho_i. \tag{1}$$

Hence, if $H < 1/2$, the auto-covariances are summable and $\mathrm{Var}(X_n)$ grows, at most, linearly fast on $n$. On the other hand, if the process has long-range dependence (LRD), $\mathrm{Var}(X_n)$ grows superlinearly on $n$. In particular, using the Euler-Maclaurin summation formula (Apostol, 1999; Alabdulmohsin, 2018), one obtains $\mathrm{Var}(X_n) \sim n^{2H}$ if $H > 1/2$.

Figure 4 plots the rescaled range $R(n)/S(n)$ against $n$. When all datasets are aggregated, we obtain an estimate of $H = 0.74 \pm .02$. As mentioned in Section 1, a value of $H \approx 0.75$ occurs commonly in nature, such as in river discharges, temperatures, and tree rings (Feller, 1951; Aref, 1998).

## 2.4 FRACTAL DIMENSION

Broadly speaking, the fractal dimension of an object describes its *local* complexity. For a geometric object $Z$, such as the Koch curve, let $\tau$ be a chosen scale (e.g. a short ruler for measuring lengths or a small square for areas). Let $N(\tau)$ be the minimum number of objects of scale $\tau$ that cover $Z$. Then, the fractal dimension of $Z$, also called its Hausdorff dimension, is (Pilgrim & Taylor, 2018): $D = -\lim_{\tau \to 0}\left\{\frac{\log N(\tau)}{\log \tau}\right\}$. For example, a line has a fractal dimension 1, in agreement with its topological dimension, because $N(\tau) = C/\tau$ for some constant $C > 0$.

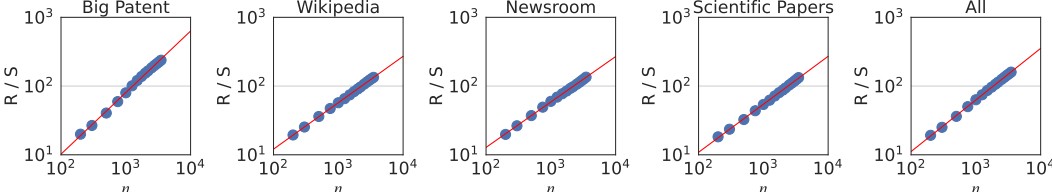

Figure 4: Rescaled range $R(n)/S(n)$ is plotted against the number of (normalized) bits $n$. We observe a power law $R(n)/S(n) \sim n^H$, with a Hurst parameter ranging from H $= 0.67 \pm .01$ in Newsroom to H $= 0.89 \pm .01$ in Big Patent. When aggregating all datasets (rightmost), H $= 0.74 \pm .02$.

Table 1: A comparison of the self-similarity exponent and the Hurst parameter on a composite of 1,000 articles sourced from both Wikipedia (Wikimedia, 2023) and Newsroom (Grusky et al., 2018) datasets. As expected, PaLM-540B produces consistently larger estimates of both parameters compared to PaLM-8B. See Section 2.5. However, both models affirm the existence of self-similarity and long-range dependence (LRD) in language.

|  | Self-Similarity Exponent S | | Hurst Parameter H | |
|---|---|---|---|---|
|  | PaLM-8B | PaLM-540B | PaLM-8B | PaLM-540B |
| Wikipedia + Newsroom | $0.60 \pm .01$ | $0.64 \pm .01$ | $0.67 \pm .01$ | $0.73 \pm .01$ |

By convention, an object is referred to as "fractal" if D is different from its topological dimension. For example, the fractal dimension of the Koch curve is about 1.26 when its topological dimension is 1. Fractals explain some puzzling observations, such as why estimates of the length of the coast of Britain varied significantly from one study to another, because lengths in fractals are scale-sensitive. Mandelbrot estimated the fractal dimension of the coast of Britain to be 1.25 (Mandelbrot, 1967).

The definition above for the fractal dimension D applies to geometric shapes, but an analogous definition has been introduced for stochastic processes. Let $(x_t)_{t \in \mathbb{R}}$ be a stationary process with autocovariance $\rho_n$. Then, its fractal dimension D is determined according to the local behavior of $\rho_n$ at the vicinity of $n = 0$, by first normalizing $(x_t)_{t \in \mathbb{R}}$ to have a zero-mean and a unit variance, and modeling $\rho_n$ using a power law $\rho_n \sim 1 - n^\alpha$ as $n \to 0^+$, for $\alpha \in (0, 2]$. Then, the fractal dimension D $\in [1, 2]$ of $(x_t)_{t \in \mathbb{R}}$ is defined by D $= 2 - \alpha/2$ (Gneiting & Schlather, 2004). A value D $\gg 1$ indicates a significant fractal structure.

It can be shown that D $= 2 - S$, where S is the self-similarity exponent (Gneiting & Schlather, 2004) so we use the latter identity in our analysis since it has the advantage of being applicable to discrete-time stochastic processes as well. For language, this gives a fractal dimension of D $= 1.37 \pm .01$.

## 2.5 ROBUSTNESS TO THE MODEL SIZE

As previously mentioned, we employed PaLM-8B for our experiments. To ensure that our main conclusions hold true even when leveraging larger LLMs, which provide more accurate estimates of the joint probability distribution in language, we calculate the self-similarity exponent and Hurst parameter using PaLM-540B (Chowdhery et al., 2022) on a composite of 1,000 articles sourced from Wikipedia and Newsroom. We then compare these outcomes with those obtained from PaLM-8B.

Table 1 summarizes the results. First, power law relations, $p_\epsilon(\tau) \sim \tau^{-S}$ and $R(n)/S(n) \sim n^H$ continue to hold when using PaLM-540B, affirming the existence of both self-similarity and long-range dependence (LRD) in language. Both exponents, however, are slightly larger than using PaLM-8B, which is quite expected. Because $H > 1/2$ quantifies predictability in language, we expect larger models to capture the predictability in language more accurately than smaller models. In addition, since the fractal dimension satisfies D $= 2 - S$, a large value of the self-similarity exponent indicates less complexity at small scales. Hence, we expect larger models to produce large estimates of S and H, in agreement with Table 1. Generally, however, these differences are not significant and we observe comparable results across both models, despite the $\times 67$ increase in model size.

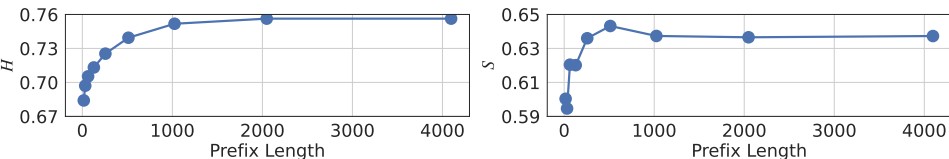

Figure 5: S and H plotted for different constructions of bits, as we vary the prefix length in the LM.

## 2.6 JOSEPH EFFECT

Finally, we also examine another related parameter that is commonly studied in self-similar processes. The motivation behind it comes from the fact that in processes with LRD, one often observes *burstiness* as shown in Figure 1; i.e. clusters over time in which the process fully resides on one side of the mean, before switching to the other. This is quite unlike random noise, for instance, where measurements are evenly distributed on both sides of the mean. The effect is often referred to as the Joseph effect, named after the biblical story of the seven fat years and seven lean years (Willinger et al., 1995; Mandelbrot & Wallis, 1968; Watkins, 2019).

A common way to quantify the Joseph effect for integral processes $(X_t)_{t \in \mathbb{N}}$ is as follows (Watkins, 2019). First, let $\sigma_\tau$ be the standard deviation of the $\tau$-increments $X_{t+\tau} - X_t$. Then, fit a power law relation $\sigma_\tau \sim \tau^J$. The exponent J here is called the Joseph exponent. In an idealized fractional Brownian motion, both J and the self-similarity exponent S coincide. Figure 6 summarizes the empirical results. When all datasets are aggregated, we obtain an estimate of $J = 0.51 \pm .01$, which is intriguing because $J = 0.5$ corresponds to self-similar processes with independent increments.

## 3 CONNECTION TO SCALING LAWS

Since the self-similarity exponent S and the Hurst parameter H quantify the level of complexity (fractal structure) and long-range dependence (LRD) in language, we expect them to correlate with the performance of language models (LMs) as we vary the context length *during inference*. To verify this hypothesis, we calculate the average log-perplexity score of PaLM-8B on each of the four datasets as we vary the length of the context window during inference in the set $\{2^4, 2^5, \dots 2^{11}\}$. Note that the model is tasked with predicting a single token only so all results are directly comparable.

Denoting $\varepsilon_x$ for the log-perplexity score, where $x$ is the context length used during inference, we model the performance as a power law: $\varepsilon_x \sim \beta x^{-c} + \varepsilon_\infty$. Figure 7 plots the scaling parameters $c$ and $\varepsilon_\infty$ in each of the four datasets against the self-similarity exponent and Hurst parameter. We observe a strong correlation in general. More specifically, a large value of S or H leads to smaller values of both $c$ and $\varepsilon_\infty$. In other words, increasing the context length offers more predictability (hence lower values of $\varepsilon_\infty$) but with a slower convergence (smaller values of $c$).

These results confirm our interpretation of the self-similarity exponent and the Hurst parameter; namely that S and H quantify the level of complexity or predictability in language. We provide further verification in Figure 5, where we plot S and H when the sequence of bits constructed using the LM are conditioned on prefixes of a small fixed length, instead of the entire history. We observe that both S and H indeed increase with prefix length (more predictability). Generally, when S and H are large, language models would benefit significantly from having longer contexts during inference.

**What about the context length at training time?**  Self-similarity and long-range dependence also point to another intriguing possibility: the importance of *training* the model with extensive contexts in order to capture the fractal-nature of language, which may elevate the model's capabilities regardless of the context length needed during inference. To test this hypothesis, we pretrain PaLM-1B (with approximately 1 billion parameters), utilizing context lengths of 256, 512, 1024, and 2048 tokens. These models are all trained under a compute-matched regime, with 12 billion tokens from the C4 corpus (Raffel et al., 2019). To assess the performance of these four models, we conduct evaluations across all tasks from BigBench-Lite benchmark. We adjust the number of shots as the main knob to control the context length during inference. The objective of these experiments is to allow us to compare the performance of models trained with different context lengths in different scenarios. For

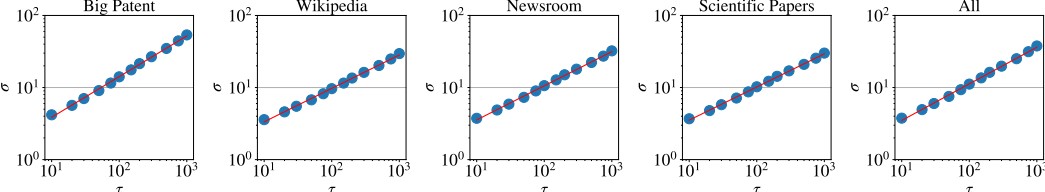

Figure 6: The standard deviation $\sigma$ of the $\tau$-increments $X_{t+\tau} - X_t$ is plotted against the scale $\tau$. We, again, observe another power law relation $\sigma \sim \tau^{\mathrm{J}}$, with a Joseph exponent ranging from $\mathrm{J} = 0.41 \pm .01$ in Wikipedia to $\mathrm{J} = 0.51 \pm .01$ in Newsroom. When aggregating all datasets (rightmost plot), we obtain the estimate $\mathrm{J} = 0.51 \pm .01$.

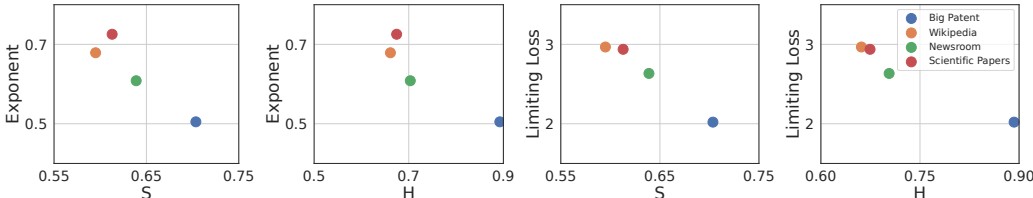

Figure 7: We model the performance of PaLM-8B on the four datasets as a power law $\varepsilon_x \sim \beta x^{-c} + \varepsilon_\infty$, where $x$ is the context length used at inference. The $y$-axis corresponds to the scaling parameters $c$ (left) and $\varepsilon_\infty$ (right) while the $x$-axis is the self-similarity exponent S and Hurst parameter H. Generally, we observe a strong correlation between fractal and scaling parameters.

instance, with small number of shots, e.g., 0 or 1, models are confronted with problems requiring shorter contexts than what they were exposed to during training. Figure 8 shows the results.

While performance turns out to be similar overall across all models, there are stark exceptions, such as `conlang_translation` and `play_dialog`, where the model trained on context length 256 tokens performed considerably worse than the rest, including in the zero-shot setting. This suggests that training a model on longer contexts might elevate its ability even if shorter contexts are used at inference time. We speculate that the reason all models perform comparably in most tasks is because C4 documents are mostly short (Xiong et al., 2022). We defer this investigation to future work.

## 4 RELATED WORKS

The statistical attributes of human language have long piqued scholarly curiosity. One example is Zipf's law, which postulates that the marginal probability (frequency) of a word follows a power law $p(\text{word}) \propto 1/r(\text{word})$, with $r(\text{word})$ symbolizing the word's rank. Shannon leveraged this observation to estimate the entropy of English to be around 1 bit per letter (Shannon, 1951), but his calculation did not consider second-order statistics. More recently, Eftekhari (2006) proposed a refinement to Zipf's law, suggesting its application to letters rather than words. Another related result is Heap's law, which states that the number of unique words in a document is a power law function of the document's length (Heaps, 1978). However, both Zipf's and Heap's laws are invariant to the permutation of words, i.e. invariant to the semantic ordering of text. Hence, they do not capture important aspects, such as long-range dependence (LRD) (Najafi & Darooneh, 2015).

In terms of self-similarity in language, the Menzerath-Altmann law stipulates a self-similar behavior in the following sense: when the size of a language construct increases, the size of its constituents decreases, and this happens at all scales (Najafi & Darooneh, 2015; Andres, 2009). In Ausloos (2012), the authors model texts as a time series by replacing a word with its length or frequency. After that, they study the fractal behavior of language. However, replacing a word with its length is invalid because it is not translation-independent (i.e. we could map every word in the language to an arbitrary token, including tokens of equal length). In our work, we model language as a time series of bits calculated from the conditional entropies, which reflects the structure of the language itself.

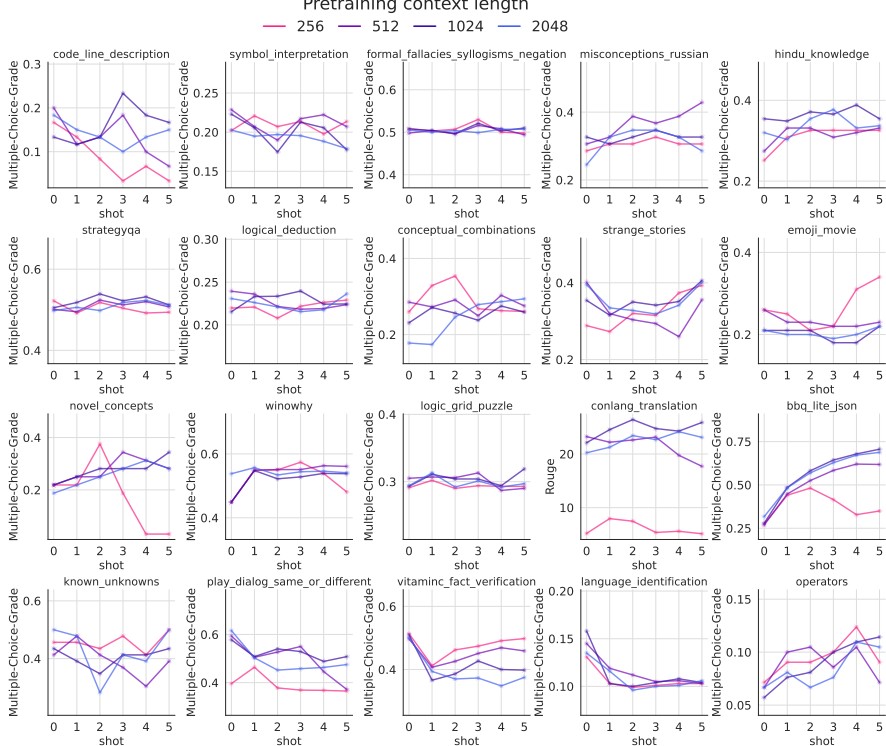

Figure 8: Performance of models trained with different context length on fewshot BigBench-Lite.

In Najafi & Darooneh (2015), the authors define a fractal dimension for each word. Informally, they examine the recurrence of a single, predetermined word in texts as an ON/OFF time series, similar to the approach used in Altmann et al. (2012). However, this is only applicable to individual words and cannot model higher-level clauses. For instance, it does not distinguish between the word "time" in the phrase "once upon a time" and the word "time" in the phrase "space and time." Kokol & Podgorelec (2000) estimate LRD in natural language texts, and suggest that LRD in language is close to that of pure noise! This raises concerns because there is clearly some dependence in language. The authors conjecture that their conclusion could be due to the use of ASCII-like encoding. In computer languages, they observe LRD and attribute this to the fact that computer languages are more formal.

Besides the above concerns in prior studies that examined the self-similar structure in language, another concern is that they sometimes give extremely large values of the fractal dimension (values that even exceed ten in some cases) (Andres, 2009). Such extreme values are difficult to interpret because classical definitions of the fractal dimension restrict its value to the range $[1, 2]$ (since the fractal dimension D is always between $d$ and $d + 1$, where $d$ is the topological dimension, which is 1 in a time series). We do not observe such issues in our analysis. In our case, $D \approx 1.37$.

## 5 CONCLUDING REMARKS

In this work, we highlight intriguing insights into the underlying fractal structure of language and how it may be interconnected with the intelligent behavior of LLMs. Our formalism quantifies properties of language that may have been suspected, but not previously formally shown. In particular, the need in LLMs to balance between short- and long-term contexts is reflected in the self-similar structure of language, while long-range dependence is quantifiable using the Hurst parameter. Interestingly, the approximate Hurst value of 0.75 for language suggests an intriguing balance between determinism and randomness that is similar to those seen in other phenomena. Also, we demonstrate how fractal patterns relate to scaling law parameters, confirming that LLMs would benefit from longer contexts in domains with a large Hurst parameter. We hope that future research can further probe into these fractal properties, unearthing deeper understandings of the relation between intelligence and language.

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
