# OpenReview forum: "Fractal Patterns May Unravel the Intelligence in Next-Token Prediction"
_ICLR.cc/2024/Conference — Submitted to ICLR 2024_

### Official Review · Reviewer_jKTM · 2023-10-30

**Soundness:** 2 fair
**Presentation:** 2 fair
**Contribution:** 1 poor
**Rating:** 1
**Confidence:** 3

**Summary:**

This paper purports to apply fractal theories to language. They claim that language is ‘self-similar’ at all levels of granularity and ‘long-range dependent’. Through a few experiments with the PaLM language model and a few disparate datasets, a few hints of self-similarity, narrowly defined, are given. A connection to scaling laws is also briefly discussed.

**Strengths:**

- It is good that four datasets were considered. It is not clear why the 4K word token cutoff is important.

**Weaknesses:**

- With regards to the supposed value of Figure 1 — a single (tripartite) anecdote showing self-similarity would not nearly be enough to show a trend, but this graph doesn’t even show unambiguous similarity — the mere presence of ‘burstiness’ is not sufficient evidence. Moreover, the Hǒlder exponent of 0.638 is claimed as sufficient to confirm self-similarity but no context is provided to contextualize this single statistic.
- The amalgamation of fractals and intelligence seems somewhat unmotivated from either an empirical or a deductive perspective.
- It seems that the authors are claiming, by means of their experimental setup, that PaLM-8B represents all of ‘language’. Later, PaLM-540B was used (in Sec 2.5) to measure ‘more accurate’ probability distributions, but wholly other architectures should have been considered and compared, too. Results should be broken down by various language families.
- The relationship between the predictability of the form that would emerge from fractal structures and those that have been approached by Zipf, Shannon, and their respective camps should be explored. I.e., despite the brief mentions of Zipfian distributions and entropy, they are not contrasted at all with the theory behind the current work; it is not clear if the authors mean to present their line of work as an alternative to well-established methods, but any deeper comparison is advised.
- Attempting to draw links between language and, say, phenomena of large bodies of water is very tenuous when the only connection is that they each share a Hurst parameter of 0.75. The authors are recommended to establish more concrete outcomes in their empirical results and conclusions, relevant to either understanding the learning process, improving outcomes, or interpreting outcomes of LLMs.

**Questions:**

- ‘Creative content creation’?
- Is it the case that LLMs <<prompted some researchers to ponder if there was more to intelligence than
“on-the-fly improvisation>>?
- you seem to have defined the $t-1^{st}$ word in terms of itself (and its prefix) in the experimental setup. You have also seemed to apply the Markov property with length $L=1$ in your definition of $z_t$. Are these true?
- By means of the $\tau$-increments, are you suggesting there are fixed, expected lengths for clauses, sentences, paragraphs (and then presumably…sections? Chapters? Articles/novels/scripts/…?).
- Could you please check your references for completeness (e.g., Aref, 1998; Bubeck et al, 2023)?

---

> ### Author Response · Authors · 2023-11-11
> **Response**
>
> Thank you for your comments and the time you have taken to review our manuscript. Please see our response below.
>
> **Figure 1**
>
> Addressing your concern about the value of Figure 1, we would like to emphasize that the purpose of this figure is to visually demonstrate what a self-similar process looks like. We never used a figure to prove self-similarity.
> The correct and well-established approach in the (extensive) literature is to show that the auto-correlecation function decays algebraically (i.e. with a power law), giving rise to the Hǒlder exponent. This is precisely what we establish! The fact that a power law holds is shown in Figure 3 (top).
>
> We elaborate on this in detail in Section 2, where we discuss the nature of self-similarity in language​​. The burstiness observed in Figure 1 is an intuitive and useful representation of self-similarity. Additionally, we provide extensive discussion on the significance and context of the Hǒlder exponent, particularly in Section 2.2, where its calculation and relevance to language are explained in detail​​.
>
> **PaLM-8B**
>
> Regarding your second point about the generalization of our findings from PaLM-8B to other models and language families, our claim is that we can obtain reasonable probability scores using LMs, such as PaLM-8B. There have been prior works that show this (we cite examples in the paper and we illustrate this calibration property in Figure 2). The reason we use PaLM-8B is merely because of the amount of compute required to conduct all the experiments.
>
> However, we did acknowledge this concern in the paper and that is precisely why we expanded our analysis to PaLM-540B, as detailed in Section 2.5 in one task. This was done to verify the robustness of our findings, and the results affirm the presence of self-similarity and long-range dependence (LRD) in language, irrespective of the model size​​. We believe this approach underscores the general applicability of our findings across different model architectures and scales.
>
> **Connection to Zipf's Law**
>
> Your suggestion to explore the connection between fractal structures in language and the works of Zipf and Shannon is discussed in Section 4. As we mention there, Zipf’s law and Shannon’s method of estimating the entropy of English only use first order statistics so they cannot be used to answer questions about long-range dependence or self-similarity. If there is anything missing that needs to be clarified, please let us know and we would be happy to elaborate.
>
> **Drawing links between language and natural phenomena**
>
> We only mention that the Hurst parameter of 0.75 is quite common in natural phenomena and the fact that it also holds for language is interesting. Regarding the relation to learning processes, we have explored this in Section 3 in which we show a connection between scaling law parameters and fractal dimensions. For example, we find that when the Hǒlder exponent and the Hurst parameter are large in a particular domain, language models would benefit from having longer contexts during inference. We also studied the connection to the context length at training time but with no conclusive results, as we discussed in Section 3 using the Big Bench benchmark. There are many other similar questions of how fractal dimensions relate to learning and we believe we are only scratching the surface here.
>
> **Answers to the Questions**
>
> * LLMs have been used to create creative content. One example is asking GPT4 to write a proof that there are infinitely many primes in the form of a poem (https://arxiv.org/pdf/2303.12712.pdf ).
> * We do provide a reference in the paper for that particular statement. It’s in Page 9 in https://arxiv.org/pdf/2303.12712.pdf.
> * In the notation, there is $w_{t-1}$, which is a single token, and there is $w_{[t-1]}$, which is the entire sequence of length t-1 tokens. So, both statements in the question are incorrect. When we query the LM to predict the probability of token $w_t$, we condition on $w_{[t-1]}$ not $w_{t-1}$. We will make the distinction easier by using boldface for $\bf w_{[t-1]}$ .
> * The lengths are random variables. Since the process is stationary, those lengths have definite expected values, but the value of course varies from clauses to sentences and sections. How the expected length changes will determine the level of self-similarity. These are standard definitions as we state in the paper and we provide references for them. We do not invent them.
> * Thank you. We will fix the missing fields in those references.
>
>
> If there are any other questions or concerns, we would appreciate it if you let us know so we can respond to them during the discussion and improve the manuscript.
>
> Thank you

---

> ### Author Response · Authors · 2023-11-17
> **Follow-up**
>
> Dear reviewer,
>
> We would like to kindly inquire if our responses have satisfactorily addressed your concerns and questions. We are open to further clarification if needed. If we have addressed your concerns, we would appreciate it if you consider revising your review/score accordingly. Otherwise, please let us know so we can respond to your inquiries during the discussion period.
>
> Sincerely

---

### Official Review · Reviewer_iZo5 · 2023-11-03

**Soundness:** 4 excellent
**Presentation:** 3 good
**Contribution:** 3 good
**Rating:** 6
**Confidence:** 4

**Summary:**

This paper aims to quantify and characterize self-similarity and long range dependency in language (token sequences) by borrowing concepts from analysis of sequential processes like Weiner process/ Brownian motion, fractal processes etc. Specifically, the natural language sequences/sentences are represented as a sequence of negative conditional logprobs (related to bits of information per token) of the tokens obtained via autoregressive language models. The actual token identities are ignored and only the conditional probability under the language model is considered. Using this representation, “increment” and “integral” processes are defined to estimate the degree of self-similarity and long-range dependency via estimating the appropriate parameters (“self-similarity exponent” and “Hurst coefficient”) via standard methods for analysing real-valued sequential processes. The language model used for this purpose is Palm-8B and 4 language corpora (Big Patent, Newsroom, Scientific papers, Wikipedia) are analyzed using the proposed method. The estimated coefficients indicate that language exhibits self-similarity i.e. it exhibits burstiness over different granualrities (sentence/paragraphs and so on), and it also exhibits long range dependency. This analysis is extended to varying model sizes and training and inference context lengths. Overall higher degree of self-similarity and long range dependence is observed as the model size and context length is increased.

**Strengths:**

– This is a fresh perspective on analysis of natural language and autoregressive language models. It provides connection to insights about the hierarchical nature of language and interconnected and recursiveness in language.

– The paper is well-written and the technique is clearly described. The approach is well-motivated and connections to prior work and techniques are illuminating.

– The experiments are well-designed. The results show robustness across a variety of domains and the relationship to scaling laws substantiates the understanding about how language models improve with scale.

**Weaknesses:**

– The identity of tokens is ignored in the analysis and the representation is just a sequence of negative logprobs. While this is still indicative of some macro features of language, this analysis is limited to an impoverished representation of language.

– As an upshot of the point above, this analysis doesn’t provide detailed insight into the exact latent hierarchical organization of language and is limited to merely an indication that language exhibits burstiness over long-range. I am unable to imagine how these concepts and methods can be extended to yield deeper insights about language and LMs beyond this paper.

– While this is a fresh perspective, the main finding that next-token prediction is a difficult task because language tends to show burstiness at varying granularities and exhibits long-range dependence is not surprising or novel. For example, the hierarchical and recursive nature of language is an actively studied area of research in cognitive sciences and linguistics.

– “Self-similarity” and “fractalness” are not identical concepts. The findings in fact suggest that language is not necessarily fractal which makes predictability challenging. This is acknowledged in the paper but yet the two concepts seem to have been used interchangeably in the discussion.

**Questions:**

N/A

---

> ### Author Response · Authors · 2023-11-11
> **Response**
>
> Thank you for your valuable insights and the careful review of our manuscript. We are grateful for your recognition of the novelty of our approach as well as your positive comments on the clarity of our writing and the design of our experiments.
>
> Addressing your first concern regarding the representation of language as a sequence of negative log probabilities (i.e. bits), we acknowledge that this approach may simplify the rich and complex nature of language. As you mention, it is unclear how the approach can be extended to take the identities of tokens into account. However, the approach can be interpreted as a lower bound to the complexity of natural language. In particular, long-range dependence on the sequence of bits implies the existence of long-range dependence on the sequence of tokens (by the data processing inequality).
>
> Regarding the concern that our findings might not be perceived as surprising or novel, we appreciate this viewpoint. While the hierarchical and recursive nature of language is a subject of study in cognitive sciences and linguistics, our work aims to bridge these concepts with the computational perspective of language models, focusing on how these inherent properties of language influence model performance. This interdisciplinary approach, we hope, adds value to the existing body of research by providing a fresh perspective to understand LLMs.
>
> We thank you for emphasizing on the distinction between “self-similarity” and "fractalness." It is not our intention to use these terms interchangeably, and we have acknowledged that they are different concepts in the paper. We will take care to avoid this in the revised version.
>
> If there are any other questions or concerns, we would appreciate it if you let us know so we can respond to them during the discussion and improve the manuscript.
>
> Thank you again for your careful and thorough critique.

---

> ### Author Response · Authors · 2023-11-17
> **Follow-up**
>
> Dear reviewer,
>
> We would like to kindly inquire if our responses have satisfactorily addressed your concerns and questions. We are open to further clarification if needed. If we have addressed your concerns, we would appreciate it if you consider revising your review/score accordingly. Otherwise, please let us know so we can respond to your inquiries during the discussion period.
>
> Sincerely

---

### Official Review · Reviewer_hBB9 · 2023-11-06

**Soundness:** 3 good
**Presentation:** 3 good
**Contribution:** 3 good
**Rating:** 6
**Confidence:** 3

**Summary:**

This paper examines the fractal structure of language and makes connections between self-similarity, long-range dependence, and the capabilities of large language models (LLMs). Fractal parameters like the Hurst exponent and self-similarity exponent are estimated on text datasets. The key ideas are that the self-similar nature of language allows LLMs to capture patterns across levels of granularity, while long-range dependence requires modeling broader contexts spanning indefinitely into the past. Experiments with PaLM on different datasets (e.g., Wikipedia, patent) empirically estimates fractal parameters like the Hurst exponent and self-similarity exponent on text datasets, finding values that suggest the presence of self-similarity and long-range dependence in language.

**Strengths:**

(+) The perspective of analyzing language through fractal patterns and quantifying self-similarity and long-range dependence provides a novel angle for understanding LLMs. While intuitions about self-similarity have existed, this paper makes them concrete and rigorous.

(+) The analyses seem technically sound overall. Estimation of fractal parameters is done carefully and situated within the literature.

(+) these insights about fractal structure connecting to capabilities of LLMs could potentially be an important conceptual advance. It formalizes useful notions about complexity and long-range dependencies in language.

**Weaknesses:**

(-) The connection between fractal structure and capabilities of LLMs is suggestive but not conclusively proven. The arguments linking self-similarity to modeling patterns across granularities seem reasonable but remain speculative without more analysis.

(-) The analysis is conducted solely on text datasets encoded using the PaLM language model. The conclusions may not generalize to other modalities or other LLMs, which is a limitation. Evaluating universality across models and data types could strengthen the results. Also, only PaLM-8B was examined, while the comparison between small and large LMs provides additional insight beyond confirming consistency. The pretraining experiment is quite brief. More in-depth studies could be beneficial.

**Questions:**

See weaknesses

---

> ### Author Response · Authors · 2023-11-11
> **Response**
>
> Thank you for your thoughtful and constructive feedback on our manuscript. We are grateful for your recognition of the novelty and technical soundness of our approach and its potential impact in advancing the conceptual understanding of large language models (LLMs).
>
> Addressing your concern about the connection between fractal structure and the capabilities of LLMs, we acknowledge that our current work is an initial step in this direction. As you suggested, extending the analysis to other modalities and models would offer interesting insights, and finding further connections between fractal dimensions and learning could uncover other results that are useful for practitioners. We plan to explore these directions in the future.
>
> In our work, we propose a precise mathematical formalism (using fractals and self-similarity) to quantify properties of language that may have been known intuitively but not formally shown, such as long-range dependence, and connected those to properties of learning. We hope this work lays the foundation for further explorations, highlighting the potential of fractal analysis as a lens for understanding and improving LLMs. We believe that additional research in this direction could provide substantial value to the field and enhance the applicability of our findings.
>
> If there are any other questions or concerns, we would appreciate it if you let us know so we can respond to them during the discussion and improve the manuscript.
>
> Thank you again for your constructive critique.

---

> ### Author Response · Authors · 2023-11-17
> **Follow-up**
>
> Dear reviewer,
>
> We would like to kindly inquire if our responses have satisfactorily addressed your concerns and questions. We are open to further clarification if needed. If we have addressed your concerns, we would appreciate it if you consider revising your review/score accordingly. Otherwise, please let us know so we can respond to your inquiries during the discussion period.
>
> Sincerely

---

### Meta-Review · Area_Chair_sYzb · 2023-12-14

**Metareview:**

This paper claims that (natural) language contains fractal patterns of self-similarity and long-range dependencies, and estimates fractal parameters of language in text data based on the conditional probabilities provided by an underlying language model. Some analyses are presented using PaLM-8B and other models under varying context lengths to justify the proposed claim.

Strengths: This paper presents a new direction in interpretability of language models, presents good analytical settings and connections to prior work on fractal processes.

Weaknesses: As reviewer iZo5 points out very clearly, this work treats language (or a model of language) as a collection of probabilities and does not take into account much beyond the burstiness of language, including hierarchical structures underlying language as dictated by an entire field of linguistics. Moreover, it is difficult to ascertain that fractal patterns alone play a part in the success of language models (exactly what is “intellligence”?), as suggested by the paper. In the absence of reasonable baselines (such as those possible under synthetic languages), it is hard to justify the statistics provided by the paper for language.

**Justification For Why Not Higher Score:**

Please see weaknesses; a somewhat shallow treatment of language as a sequence of conditional probabilities is equated to the “intelligence” of language models.

**Justification For Why Not Lower Score:**

N/A

---

### Decision · Program_Chairs · 2024-01-16

Reject